# Experiences and Lessons Learned from COVID-19 Pandemic Management in South Korea and the V4 Countries

**DOI:** 10.3390/tropicalmed6040201

**Published:** 2021-11-25

**Authors:** Gergő Túri, Attila Virág

**Affiliations:** Institute for the Development of Enterprises, Corvinus University of Budapest, Fővám Square 8, 1093 Budapest, Hungary; gergo.turi2@stud.uni-corvinus.hu

**Keywords:** COVID-19, South Korea, V4 countries, Czech Republic, Hungary, Poland, Slovakia, pandemic management, health policy, research and development

## Abstract

In the first year and a half of the COVID-19 pandemic, South Korea suffered significantly less social and economic damage than the V4 countries (Czech Republic, Hungary, Poland, and Slovakia) despite less stringent restrictive measures. In order to explore the causes of the phenomenon, we examined the public health policies and pandemic management of South Korea and the V4 countries and the social and economic outcomes of the measures. We identified the key factors that contributed to successful public health policies and pandemic management in South Korea by reviewing the international literature. Based on the analysis results, South Korea successfully managed the COVID-19 pandemic thanks to the appropriate combination of non-pharmaceutical measures and its advanced public health system. An important lesson for the V4 countries is that successful pandemic management requires a well-functioning surveillance system, a comprehensive testing strategy, an innovative contact tracing system, transparent government communication, and a coordinated public health system. In addition, to develop pandemic management capabilities and capacities in the V4 countries, continuous training of public health human resources, support for knowledge exchange, encouragement of research on communicable disease management, and collaboration with for-profit and non-governmental organizations are recommended.

## 1. Introduction

The COVID-19 pandemic is one of the biggest challenges of today, requiring significant efforts from all governments. In China, the emergence of coronavirus (SARS-CoV-2), which causes a severe respiratory syndrome, was identified in December 2019 [1]. In January 2021, the virus had already appeared in neighboring countries, including South Korea, and by March, the first infected people had been identified in Europe, including the V4 countries [2]. The WHO declared a pandemic on 11 March 2020, when the virus was already present in 114 countries [3]. Since then, as of 10 November 2021, approximately 250 million confirmed COVID-19 cases and five million deaths have been reported worldwide in connection with SARS-CoV-2 infection [4]. According to a recent study analyzing the geographical distribution of the pandemic, the highest cumulative totals per 100,000 population confirmed cases were registered in Europe (49.3%) and the Americas (27.4%) by July 2021, while the lowest was in Southeast Asia (2.7%) [5].

Governments had to deal simultaneously with the health emergency threatening the lives and quality of life of the population and the adverse economic effects of measures to slow the spread of the pandemic [6]. 

The pandemic also caused severe problems in other areas: distance learning may have increased the backlog in education in disadvantaged social groups, mental health problems have increased in many countries, the production and handling of protective equipment have often caused a severe environmental burden, and pandemic measures have often led to significant social dissatisfaction [7]. To address the complex, multi-factor problem presented by COVID-19, governments have applied different strategies, leading to different health and economic outcomes. The outcomes of the year and a half since the pandemic provide an opportunity to assess different countries’ experiences in managing the pandemic so far. 

Our analysis compares and evaluates the public health policies and pandemic management of South Korea and the V4 countries for the COVID-19 pandemic. The issue is relevant because South Korea has suffered significantly less social and economic damage than the V4 countries despite applying less stringent restrictive measures. Therefore, in this analysis, we examine the following questions: How can the pandemic management strategies of South Korea and the V4 countries be characterized? What led to the health and economic outcomes of different strategies? Why has South Korea suffered fewer health and economic losses than the V4 countries? In order to better manage the COVID-19 pandemic and future pandemics, in which areas should V4 countries develop their capabilities and capacities? 

If the pandemic is interpreted as a complex system that behaves unpredictably, it becomes clear that managing it requires complex decision-making and monitoring systems, along with novel management systems and mechanisms that require a wide range of expertise [7,8]. The novelty of our analysis is that by comparing the pandemic management of the V4 countries and South Korea, we reveal key elements of these complex systems that will allow the governments and societies of the V4 countries to achieve both low health and economic losses at the same time.

## 2. Materials and Methods

In our analysis, we first summarize and evaluate the study’s theoretical frameworks while describing the key concepts used in the research. The governmental actions, public health policies, and epidemiological management strategies of the examined countries are described using data from the international literature and databases. In summary, we highlight the similarities and differences in pandemic management in the V4 countries and the specifics of pandemic management in South Korea. We then review and evaluate the health and economic achievements of each country. 

In exploring the reasons for successful pandemic management in South Korea, we present the primary theoretical approach to the country’s epidemiological management strategy. We also review which innovative organizational, management, and technological solutions were used and how governmental, economic, and various social actors worked together. Based on the lessons learned from the South Korean example, we then summarize the areas in which the V4 countries should develop their pandemic management capabilities and capacities, identifying areas where R&D resources should be focused in the future.

### 2.1. Theoretical Background

In the theoretical background of the analysis, we describe and evaluate two theoretical frameworks. While the first framework contains a system of tasks necessary for the successful management of pandemics, the second concept defines the main functions of public health systems. Although several additional concepts and frameworks are available for assessing pandemic management [9,10], we have chosen the two frameworks described below for our analysis because they provide a wide range of perspectives.

#### 2.1.1. A System of Tasks Necessary for the Successful Management of a Pandemic

Successful management of a pandemic requires, on the one hand, pharmacological measures such as effective and efficient antiviral drugs or vaccines [11,12]. Non-pharmaceutical measures, which can be grouped into the following three main categories, are often also indispensable tools for successful management: 

(a) Reducing the number of physical contacts made (e.g., curfew restrictions, restrictions on group gatherings, closure of nightclubs and restaurants, restrictions on domestic and international travel).

(b) Reducing the transmission of infection through individual-level precautions (e.g., mandatory mask use indoors and outdoors, application of hand and respiratory hygiene rules).

(c) Controlling the source of infection (e.g., extensive testing, contact research, quarantine of persons confirmed to be infected and individuals likely to be infected).

On the emergence and rapid spread of a new respiratory virus in the community, non-pharmacological measures may be essential if antiviral drugs and vaccines are not yet widely available. Although non-pharmaceutical measures can have severe health and economic consequences, a combination of measures appropriate to the epidemiological situation can effectively reduce the rate of spread of the virus. Due to the flattening of the pandemic curve, fewer new infections occur later, resulting in a lower burden on the health system and more moderate health effects than not introducing restrictive measures. However, the management of the pandemic should not be limited to the application of these measures. It is necessary to ensure there is adequate health capacity to treat infected citizens and implement vaccine development and vaccination programs. With non-pharmaceutical measures, time can be gained for vaccine development while also reducing health losses.

#### 2.1.2. Ten Essential Public Health Operations (EPHOs) of a Public Health System

A well-functioning public health system needs to perform several functions and tasks when dealing with a pandemic. The framework established by the World Health Organization (WHO) sets out the operations for protecting and improving the population’s health and preventing disease [13]. The tasks detailed in the concept cover a much more comprehensive range than is necessary for the prevention and management of health emergencies, but all tasks (and some of their elements) contribute directly or indirectly to the successful management of pandemics. The first two basic public health tasks are monitoring the health of the population and monitoring the factors that affect their health. These tasks include collecting information on the spread of non-communicable and communicable diseases in various databases and surveillance systems. 

The first two EPHOs provide essential information for planning, modelling, and evaluating public health interventions. During health emergencies, data collected in surveillance systems can serve as the input, for example, in models that predict the spread of a pandemic, based on the results of which a government can take action [14]. The third EPHO is health protection, which, in addition to food safety and occupational health activities, also includes the development of rule systems that ensure the protection of the population’s health and the observance of the rules. In health emergencies, a clear focus is on the straightforward design, application, and monitoring of epidemiological rules. 

The aim of the health promotion classified in the fourth EPHO is to develop the population’s health behavior and culture, so that individuals positively influence the factors affecting their health. Health promotion can be considered a key element to the COVID-19 pandemic because the course of the disease is negatively affected by several chronic diseases and risk factors [15,16]. Thus, if members of a social group are characterized by poor health indicators and widespread risk factors, there is a significant increase in the likelihood that they will face greater social and economic losses as the pandemic spreads. 

The fifth EPHO is disease prevention and early detection among both communicable and non-communicable diseases. During health emergencies, the task is to apply extensive screening tests and reduce health risks by using vaccines. The sixth EPHO is the management of the public health system, which should operate public health organizations and actors at the national, regional, and local levels in a planned manner with a defined strategic goal system, coordinating their tasks. 

The seventh EPHO is to provide public health human resources with the proper knowledge and experience to carry out all tasks. In the event of a health emergency, the availability of epidemiologists, analysts, and contact researchers with extensive knowledge and experience, and if necessary, further development of human health human capacities, are of paramount importance. The eighth EPHO is the adequate financing of the public health system, which will ensure its sustainable operation in the long run. 

The ninth EPHO is health communication aimed at all population groups and various sectoral actors and organizations. In the event of health emergencies, health communication can be attributed to playing a key role in influencing the health behavior of the population. The tenth EPHO is to promote and implement public health research and development, use research results in decision-making, and develop public health programs. Research and development are essential for the successful management of pandemics, enabling infectious diseases to be diagnosed, treated, and cured as accurately and quickly as possible or prevented by vaccination. 

The successful management of the COVID-19 pandemic requires complex and multifaceted measures and a well-functioning public health system [7,8]. Various non-pharmaceutical measures can be used to slow down the spread of the pandemic, but they can also have significant social and economic disadvantages. Another set of tools needed to successfully manage a pandemic are pharmacological measures (effective and efficient health therapies, along with vaccines that can reduce the risk of the disease). However, developing these therapies and vaccines is a time-consuming task; countries can gain the necessary time by using non-pharmacological measures. Well-functioning public health systems provide the knowledge, tools, and resources needed to successfully manage a pandemic. Therefore, it is crucial that each country develops its public health systems before the pandemic, providing adequate monitoring, methodological, and analytical capacities, along with human resources, regulatory, funding, and communication systems. It is also essential to develop the R&D&I infrastructure, carry out research, and integrate the results into daily practice.

### 2.2. Source of the Data and the Methods of Data Analysis

We used information and indicators from the ‘Our World in Data’ (OWID) database, operated by the University of Oxford, to examine the public health policies and pandemic management of South Korea and the V4 countries [4]. In evaluating the non-pharmaceutical measures taken to reduce physical contact, we used the stringency index developed by the University of Oxford, which consists of nine components: closing schools, closing jobs, postponing community events, restricting gatherings, public transport closure, stay-at-home measures, communication campaign, international travel restrictions, and international travel controls [4]. Information on face-covering policies and testing strategies was also evaluated during the evaluation of non-pharmaceutical measures. Vaccination data in V4 countries and South Korea were reviewed during the examination of pharmacological measures.

During the data collection and analysis, the period from 1 January 2020 to 31 March 2021 was examined. For some indicators and topics, the time horizon was divided into two: 1 January 2020–6 September 2020; and 6 September 2020 to 31 March 2021. The two time-bands of the analysis were split into two parts on 6 September 2020 because the onset of the second wave of the pandemic in the V4 countries can be identified as having begun around August–September 2020. For certain indicators, we used a shorter time interval (such as the change in vaccination rates over time, from 1 January 2021–31 March 2021), while for some indicators, we evaluated and compared information for specific one-to-one timepoints. The examined indicators and information were grouped according to the two main measures formulated in the theoretical background (pharmacological and non-pharmaceutical measures). 

Data and information from the OWID and OECD databases were used to examine the pandemic management’s health and economic outcomes in South Korea and the V4 countries. The following indicators were examined during the evaluation of the health and economic outcomes: seven-day moving average of new cases per one million population per day, cumulative number of confirmed COVID-19 deaths per one million population, weekly excess mortality, quarterly GDP change year-on-year, and quarter-on-quarter change in GDP. During the collection of data on health outcomes and the preparation of the analysis, we examined the period from 1 January 2020 to 31 March 2021. Similar to the review of public health policies and pandemic management, for several indicators and themes, the time horizon was divided into two parts: 1 January 2020–06 September 2020 and 6 September 2020 to 31 March 2021. For some indicators, we examined the results achieved during the entire period under review. 

To identify key factors in South Korea’s successful pandemic management, we conducted literature collection with the following keywords in the PubMed, Scopus, Web of Science, Science Direct, and Google Scholar databases: South Korea, COVID-19, public health system, pandemic management, health emergency management. The inclusion criteria for the publications identified during the literature search were as follows: the publication should include information on the operation of the country’s public health system, the functions and responsibilities of pandemic management organizations, and experience in managing the COVID-19 pandemic.

Exclusion criteria were publications more than five years old and publications that did not contain good practices related to pandemic management. While processing the literature, we focused on understanding the tools and methods that helped South Korea to achieve lower health and economic losses than the V4 countries, despite the less stringent restrictive measures. In identifying areas for improvement for more successful public health policies and pandemic management in the V4 countries, we examined the experiences and lessons learned from earlier topics. Based on the analysis, we formulated proposals for the areas of the public health systems of the V4 countries in need of development, as well as their pandemic management strategies and tools.

## 3. Results

### 3.1. Public Health Policies and Pandemic Management in South Korea and the V4 Countries

#### 3.1.1. Examination of Non-Pharmaceutical Measures

The severity of restrictive measures in each country is indicated by the stringency index with values between 0 and 100, with ‘0’ being the least stringent and a value of ‘100’ being the most stringent. In parallel with the comparison of the stringency index values, it is expedient to analyze other epidemiological indicators of the examined countries. It is only in the context of these that it is possible to assess whether a government has applied measures at the right time, in the correct number, and with suitable rigor. The stringency index values of South Korea and the V4 countries (excluding Hungary) started to increase in January 2020 (Figure 1). In South Korea, the first case was identified on 24 January 2020, and progressively more stringent measures were taken over the next 10 days (stringency index value: 30) [4]. First cases were identified in the V4 countries between 1 and 6 March 2020, and the stringency index values reached 30 in 6–10 days.

While in South Korea, the period of strict restrictions was characterized by a stringency index above 50, which lasted for two months (23 February to 20 April), in the V4 countries, this period lasted three months (11 March to 12 June). Between June and August 2020, the Czech Republic, Slovakia, and Poland applied less stringent measures than South Korea, even though the new confirmed cases each day per one million people in these countries were between 2 and 20 times higher than in South Korea [4].

Between 22 October 2020 and 31 March 2021, the V4 countries applied significantly stricter restrictions than in South Korea (Figure 2). When the V4 countries reintroduced strict restrictions between 22 October and 1 November 2020 (stringency index values were above 50), the daily new confirmed case per one million people were 77–520 times higher in these countries than in South Korea [4]. While the stringency index values of the V4 countries were mainly in the range of 70–80 between October 2020 and March 2021, in South Korea, except for a total of 20 days, the index value remained below 65. 

One of the essential measures to reduce the transmission of infection is the use of a mask. Based on the OWID database, the government in South Korea was the first of the countries to state that mask-wearing was recommended as of 16 March 2020 (Table 1). The governments of the V4 countries made wearing masks mandatory in certain public places between 19 March and 28 April 2020. Nearly half a year later, two countries (Czech Republic and Slovakia) further tightened of the rules on mask use. 

An essential element of measures to control the source of infection is testing practice. South Korea was the first country in the world to implement a comprehensive testing policy, on 7 February 2020, which included both symptomatic and asymptomatic individuals (Table 2). This testing strategy allowed South Korea to identify as many infected people as possible in the pandemic’s early stages, breaking the chains of infection. The V4 countries did not develop their testing policies until a month later, in mid-March, targeting only symptomatic individuals who also met one of the additional criteria, such as close contact with a confirmed infected person, a healthcare worker, or someone from abroad who had arrived home. Only Slovakia and the Czech Republic, of the V4 countries, developed the same comprehensive testing strategy as South Korea (in September and December 2020, respectively). In the emerging phase of a pandemic, the development of testing capacities and extensive testing programs is essential due to the high rate of asymptomatic infection. International studies indicate that at least one-third of SARS-CoV-2-infected people are asymptomatic, and transmission from asymptomatic individuals accounts for more than half (59%) of all infections [17,18]. 

Another critical indicator of the testing strategy is the number of tests per confirmed COVID-19 case. According to the WHO recommendation, the ideal level is 10–30 tests per confirmed case [19]. A lower value may mean that the extent of testing does not follow the spread of the pandemic, so surveillance systems do not notice many cases. At the outbreak of the pandemic, South Korea began very intensive testing, with nearly 1800 completed tests at the peak for one confirmed COVID-19 case (Figure 3). Until 6 September, this value was consistently maintained above 30, which means that extensive testing was maintained. A similar picture emerges for the V4 countries, which significantly expanded their testing capacities as recommended by the WHO, carrying out more than 30 tests for one confirmed case. 

However, a completely different trend can be observed when looking at 6 September 2020 to 31 March 2021 (Figure 4). The number of tests per confirmed COVID-19 case shows that from September onwards, the testing rate in the V4 countries did not follow the rate of spread of the pandemic and remained below the WHO recommendation until January 2021. From January 2021 in Slovakia, and from the beginning of March in the Czech Republic, the testing pace increased, while Hungary and Poland still did not carry out testing to the recommended extent. In contrast, South Korea maintained extensive testing throughout the period under review. 

#### 3.1.2. Examination of Pharmacological Measures

Safe and effective vaccines are essential tools for the management of the COVID-19 pandemic. Thanks to the joint procurement of vaccines in the European Union, the V4 countries were able to launch their vaccination programs from January 2021, as a result of which 11–21% of their population received at least one dose of vaccination, equating to 6–12 times the South Korean level of cover (Figure 5). The difference in the Hungarian vaccination rate compared to the other V4 countries arose because, in addition to the joint procurement in the EU, the country also uses vaccines from the Russian Sputnik and Chinese Sinopharm vaccine manufacturers in its vaccination program. South Korea began its vaccination program nearly two months later, in late February 2021, and by the end of March, the proportion of the population vaccinated with at least one dose had increased but at a slower rate. 

### 3.2. Health and Economic Achievements in South Korea and the V4 Countries

#### 3.2.1. Results of Measures to Slow down the Spread of the Pandemic

A seven-day moving average of new cases per one million people per day can be a helpful indicator when examining the topic. Although this indicator well characterizes the different magnitudes of pandemic waves between countries, its limitation is that screening practices in each country can significantly influence how many infected people are found, so limited conclusions can be drawn when comparing data. In the V4 countries, despite less extensive testing, a notably higher number of infected people were diagnosed between January and September 2020 than in South Korea (Figure 6). In Hungary and the Czech Republic, the numbers of cases at the beginning of September were already 7–10 times higher than the pandemic curve in South Korea.

Thus, between 6 September 2020 and 31 March 2021, governments in the V4 countries could not slow down the spread of the pandemic, or only to a limited extent. It is noteworthy that while in South Korea, the seven-day moving average of new cases per million population ranged from 1 to 20 between September 2020 and March 2021, in the V4 countries, despite more limited testing practices, there were 4 to 600 times the number of new cases compared to South Korea (Figure 7). 

#### 3.2.2. Health Outcomes

An important indicator of health outcomes in pandemic management could be the cumulative number of confirmed COVID-19 deaths per one million population, but comparison between countries is limited. The number of COVID-19 deaths may be affected by different country-specific definitions of COVID-19 deaths, possible administrative/data recording errors and delays, and the extent of testing. 

Two different methodologies are most commonly used to define deaths: in some countries, COVID-19 mortality is the death of a test-confirmed COVID-19 case that resulted in a clinically compatible disease with COVID-19 infection [20]. In contrast, some countries classify those confirmed to be infected with COVID-19 and those who may have died from COVID-19 infection based on their previous symptoms, test results, and death certificate.

Finally, concerning the definitions of death, if a country determines COVID-19 mortality statistics solely based on confirmed cases, mortality may be significantly lower than it is if testing capacity remains low, despite the wide spread of the pandemic. Between 6 September 2020 and 31 March 2021, the V4 countries were affected by several large-scale pandemic waves. As a result, by the end of the study period, the cumulative number of COVID-19 deaths per one million population in these countries was 41–73 times that of South Korea (Figure 8). 

Another widely used indicator of the impact of the COVID-19 pandemic is the rate of excess weekly mortality compared to the average of the previous five years. As this indicator is based on all deaths in a given period, problems with the COVID-19 death definitions can be eliminated. While there were 5–10% excess deaths in South Korea between January and March 2020, there were 5–10% fewer deaths in the V4 countries compared to the average of the previous five years. Between September 2020 and 31 March 2021, the V4 countries had significant excess mortality compared to South Korea (Figure 9).

While more than 100% excess mortality can be observed to have occurred in the V4 countries, in South Korea, this indicator ranged from −5% to 9% during the period under review. In their analysis published in 2020, Bogos et al. mention that the V4 countries’ excess mortality data for the fourth quarter proved to be remarkably high [21]. According to an analysis by Barański et al., there was significant excess death in Poland in 2020 [22]. Based on the evaluation of the health results related to the management of the pandemic, the V4 countries applied more stringent measures and for a more extended period, but nevertheless suffered more substantial health losses when compared to South Korea.

#### 3.2.3. Economic Outcomes

One of the indicators that can be used to assess the economic results related to the management of the pandemic is the percentage change in quarterly GDP compared to the same quarter of the previous year. According to OECD data, the V4 countries recorded a more notable decline in GDP of three to four times greater than South Korea in the second quarter of 2020 (Figure 10). 

In the third and fourth quarters of 2020, this discrepancy and the extent of the discrepancy remained similar (two to four times different). In the first quarter of 2021, while there was a notable increase in GDP in South Korea, stagnation or a decrease in GDP was observed in the V4 countries. The economies of the V4 countries were more severely affected by the COVID-19 pandemic than South Korea. 

A similar finding was observed when examining the percentage change in quarterly GDP compared to the previous quarter (Figure 11). In the second quarter of 2020, the V4 countries recorded a 2.2–4.3-fold decline in GDP compared to South Korea. 

In Q3 2020, however, due to the easing of restrictions, GDP in the V4 countries grew more than in South Korea. In Q4, except for Poland, all countries produced GDP growth close to the OECD average. Then, in the first quarter of 2021 in South Korea, Hungary, and Poland, the economy grew above the OECD average compared to the previous quarter. Based on assessment of the economic results related to the management of the pandemic, the V4 countries applied stricter restrictive measures for a more extended period than South Korea in Q2 2020, and therefore, suffered more substantial economic losses. The economic performance of the V4 countries was then hindered by the fact that in Q4 2020, they had to apply wide-ranging restrictions due to the re-emergence of the pandemic.

### 3.3. Key Factors in South Korea’s Successful Public Health Policies and Pandemic Management

#### 3.3.1. Key Factors Based on the WHO Framework Categories

An exciting question arises as to why South Korea suffered fewer health and economic losses than the V4 countries since the COVID-19 pandemic began in January 2020. The background to the response is the MERS (i.e., Middle East respiratory syndrome) pandemic in South Korea five years earlier, which caused the country severe economic losses [24]. During the three months of the 2015 MERS pandemic in South Korea, 186 confirmed cases of infection were identified, and several thousand individuals had to be quarantined [25]. Following the 2015 MERS pandemic, the South Korean government implemented nearly 50 reforms to prepare the country for more effective management of future pandemics [26]. In this section of the article, structured according to the WHO framework described in the theoretical background, we describe the key factors in successful South Korean pandemic management (Table 3). 

##### (EPHO 1 + 2) Monitoring

Following the MERS pandemic in 2015, the Ministry of Health and Welfare passed several reforms in public health [40]. Based on experience of the MERS pandemic, the government established the Emergency Operations Center, which monitors infectious diseases in real-time, 24 h a day, seven days a week [26,27]. In addition to monitoring epidemiologically risky cases, the organization’s task is to identify possible pandemic outbreaks, assess such a situation, and if necessary, immediately implement epidemiological measures commensurate with the risks [29]. The center regularly conducts outbreak simulations and develops risk-management strategies, in addition to the ongoing evaluation of data and information from South Korean surveillance systems [28].

##### (EPHO 3) Health Protection

The Communicable Diseases Control and Prevention Act 2015 sets out in detail the roles and responsibilities of various governmental organizations in the event of a health emergency. The legislation also provides for compliance with epidemiological rules, with the possibility of fines for non-compliance [31]. In addition, the South Korean government has developed a regulatory system that allows the Korean authorities to assess and authorize the use and manufacture of newly developed testing devices in as little as one to two weeks, instead of requiring several months, in an emergency [32]. In South Korea, social distance levels appropriate to the pandemic situation have been formulated. As the number of new cases per day increases, the distance rules also become stricter, with epidemiological organizations constantly informing the population groups of what to do. 

##### (EPHO 4) Health Promotion

In South Korea, nationwide health promotion activities and programs are funded by the health promotion fund, which receives tax revenues from the sale of tobacco products [34]. The healthy cities program has been operating in the country since 2004. The aim is to develop and implement health-promoting policies and health plans, support the creation of a healthy environment, and encourage the population to lead a healthy lifestyle [35]. According to an OECD report, South Korea also places a significant focus on health-promotion strategies and programs to prevent smoking or excessive alcohol consumption, as well as support healthy eating and physical activity among young people [36]. 

##### (EPHO 5) Disease Prevention

In South Korea, aggressive and widespread testing began after the first case appeared, involving hundreds of testing centers in healthcare facilities, as well as laboratories and drive-through centers [29]. Temporary screening stations have been set up in public spaces in front of hospitals to prevent the formation of hospital focal points. South Korea has made a unique development in the field of contact tracing. The officers of the Pandemic Intelligence Service (EIS) are responsible for carrying out this task [29]. If someone’s test results confirm SARS-CoV-2 infection, EIS professionals will personally interview them in the first phase of contact research (Discovery) about where they were in the days before and with whom they came in close contact [39]. In the first phase, EIS professionals can also interview close family members and the treating physician of the infected person. In the second phase of the contact tracing (Risk Assessment), the EIS specialists check the data and information provided by the infected person (and their family members and doctor) during the interview. To do so, contact researchers can access GPS data on an individual’s cell phone, location data for their credit card purchases, patient data stored at healthcare providers, and public camera recordings stored by the police. In the third phase of contact research, contacts are classified based on the data collected during the risk assessment. The fourth stage is contact management, during which close contacts are quarantined and average contacts are further monitored as needed. At the beginning of the COVID-19 pandemic, the South Korean government transformed the workers’ hostels, dormitories, and leisure facilities of many private companies into temporary isolation facilities [29]. The government’s goal was to reduce the pressure on hospitals, and this way, quarantined individuals in facilities did not infect their family members either. 

##### (EPHO 6) Governance

Due to difficulties in cooperation between government agencies and private services at different levels (national and local) during the 2015 MERS pandemic, in the aftermath, the South Korean government clarified legislation on the division of tasks and cooperation between these actors [26]. Health emergencies are now managed via a decentralized system [29].

##### (EPHO 7) Public Health Workforce

The Korean government has placed great emphasis on the development of human resources for epidemiology. The objectives of the reform launched in 2015 included improving the number and knowledge of epidemiologists in the country. Supporting their international knowledge and exchange of experiences was a further goal, which allowed Korean professionals to learn about and master epidemiological systems in the years that followed, especially in China and the United States [40]. During the COVID-19 pandemic, the government significantly increased the country’s number of public health professionals, involving both private professionals and volunteers [29].

##### (EPHO 8) Funding

The government provides the resources required for the proper functioning and development of the public health system under the Communicable Diseases Control and Prevention Act 2015 [26]. Thanks to that regulation, COVID-19 tests and health services were widely available and free of charge for the population, thus ensuring an adequate level of participation in testing by lower-income social groups. The regulation allows for the mobilization and allocation of additional resources to the budget in a health emergency [32]. 

##### (EPHO 9) Communication

During the COVID-19 pandemic, the Office of Risk Communication, established by the KCDC, was responsible for wide-ranging and transparent communication to all sections of the population [26]. The agency provided current information and regulations about the pandemic situation on the Internet, radio, and television channels, tailored to the target group, and drew the public’s attention to social distancing and compliance with hygiene standards. The KCDC and government agencies regularly provide detailed data on and analysis of the evolution of the pandemic situation [32]. 

##### (EPHO 10) Research

In the years following the MERS pandemic, several biotechnology companies emerged that played an essential role in the COVID-19 pandemic. Even before the COVID-19 pandemic, these companies worked with government organizations in several PPP collaborations [26]. Following the appearance of the first case in South Korea in January 2020, the KCDC asked biotechnology companies operating in the country to develop and manufacture diagnostic tools and reagents [32]. Further to this, testing companies produced tens of thousands of tests per day for weeks, which allowed for a rapid and continuous increase in the number of tests performed, supported by the operation of hundreds of test centers, laboratories, and drive-through stations nationwide [29]. 

#### 3.3.2. The Cultural and Social Background of South Korean Pandemic Management

The successful management of pandemics can be significantly influenced by how much the country’s citizens trust the government. In South Korea, trust in the government has been at a markedly low level for decades [41]. Compared to the V4 countries’ confidence index values, we found similar values in South Korea (Table 4), so the question arises as to what encourages South Korean citizens to cooperate and comply with epidemiological rules. 

The response can be linked to the 2015 MERS pandemic, as in many previous areas. According to a 2019 analysis by Lee and colleagues, factors that greatly influenced the management of the MERS pandemic included transparent and wide-ranging public communication, information, and information sharing [30]. The majority of the public trusts the contact research system, despite raising issues of privacy rights in some areas. During the COVID-19 pandemic, the South Korean government led extensive communication and information campaigns targeting various population groups. According to a recently published survey, transparent and clear government communication and successful pandemic management have greatly increased trust in the government, though confidence in religious groups, which have been linked to several outbreaks, has declined significantly [43]. According to the OECD, trust in the government increased in South Korea in 2020, while among the V4 countries, trust in the Czech Republic, Poland, and Slovakia deteriorated notably, with only a small increase in Hungary (Table 4).

There is also a social background of South Korea’s successful pandemic management, based on cooperation between the government and for-profit companies. Examples include biotechnology companies that played a prominent role in developing and producing tests and reagents. Based on the literature review results, it can be understood that the South Korean government has established many PPP collaborations with private companies in the field of health, promoting research into and development of diagnostic tools and therapies for infectious diseases.

### 3.4. Areas for Improvement for More Successful Public Health Policies and Pandemic Management in the V4 Countries

Based on the experience and lessons learned from pandemic management in South Korea, several suggestions can be made as to which areas V4 countries should develop their capabilities and capacities in for more successful pandemic management. In order to be able to localize pandemics in space and time and take targeted and risk-proportionate measures, well-functioning surveillance systems are needed, consisting of several subsystems. Therefore, it is appropriate to continuously improve existing surveillance systems to ensure the monitoring of factors affecting the health of the population and the monitoring and evaluation of the effects of measures. 

Well-functioning surveillance systems consist of several subsystems that provide essential information for monitoring and successfully managing a pandemic [44]. The continued emergence of newer variants of coronavirus makes the establishment of surveillance systems particularly important. They can be used in time to identify, for example, whether the protective effects of vaccines are holding for a new variant. Further to this, in order to successfully manage health emergencies, it is appropriate to coordinate the tasks and functions of organizations operating at the national and local levels and provide resources for the operation and development of the public health system. Moreover, it is recommended for the V4 countries to expand health promotion services and capacities to improve the population’s health behavior and health culture. The development of public health programs aimed at healthy eating, physical activity, preventing smoking and excessive alcohol consumption, and creating a healthy environment can bring significant social and economic benefits, as the risk factors for more serious diseases can be reduced even in the short term. 

Beyond this, it is recommended that the V4 countries adopt a comprehensive testing strategy tailored to the pandemic situation, enabling access for all social groups. By applying a contact research system and isolation procedures using innovative technological solutions, a significant portion of the infection chains can be interrupted, and the load on human health resources in public health can be reduced. Although complete rollout of the contact research system in South Korea is likely to provoke significant resentment in some social groups, the contact research system in the V4 countries should be further developed, as the South Korean example shows that timely detection and disruption of infection chains can significantly slow down the spread of the pandemic. 

It is also recommended that the governments of the V4 countries continuously improve their health human resource capacity. In addition to increasing the number of public health professionals, it is expedient to support training, scholarship programs, and international knowledge exchange programs that ensure professional development and staying on track. Proposed research areas include the development of therapies for the treatment of infectious diseases and the development and production of vaccines. In all areas, emphasis should be placed on developing collaborations between government, for-profit, and non-governmental organizations and on using incentive schemes that promote research and innovation in the prevention, treatment, and control of communicable diseases.

## 4. Conclusions

To the best of our knowledge, this is the first research to compare the public health policies and pandemic management experiences of South Korea and the V4 countries.

With our first research question, we sought to answer how to characterize the public health policies and pandemic management of South Korea and the V4 countries. Based on the research results, we understand South Korea to have implemented an extensive testing program from the initial stage of the pandemic, coupled with innovative contact research and isolation methods. In the early stages of the pandemic, a testing strategy was used that targeted both symptomatic and asymptomatic individuals. Restrictive measures in the country were gradually tightened in the initial phase of the pandemic. However, more stringent measures were only in force for two months, after which they were significantly eased, and no national closures were required during the period under review. 

In contrast, in the initial phase of the pandemic, the V4 countries, at almost the same time, took stringent measures at a rapid pace, which were maintained for a month longer than in South Korea. Their testing strategy in the first half of the pandemic was less extensive, so cases could escape the attention of the epidemiologist. Since the September 2020 phase of the pandemic, there have been severe pandemic waves in the V4 countries. Based on the data, these countries did not carry out sufficient tests, so the contact researcher could not break all infection chains. Because of this, many severe restrictions and national closures were needed in October 2020 and March 2021 in the V4 countries. Testing was carried out in Slovakia from January 2021 and the Czech Republic from March 2021 to a degree that is in line with WHO recommendations, but in addition to testing, the development of contact research and isolation procedures and systems is essential. In the field of vaccinations, the V4 countries gained a significant advantage over South Korea, with Hungary achieving ten times the coverage level of South Korea by 31 March 2021. 

Our second research question concerned the health and economic outcomes of the different strategies in the countries studied. Based on the analysis, the cumulative numbers of COVID-19 deaths per one million population in the V4 countries were 41–73 times greater than the South Korean mortality rate as of 31 March 2021. A similarly significant divergence was observed in the area of excess mortality, which affected the V4 countries more severely than South Korea. Overall, the V4 countries suffered more significant health losses despite stricter restrictions and national closures being in place for a significant part of the period under review, while this was not necessary for South Korea. In terms of economic performance, there were also greater losses in the V4 countries, with a larger quarter-on-quarter decline in GDP than South Korea when compared to the same period last year. 

With our third research question, we sought to answer why South Korea suffered lesser health and economic losses than the V4 countries. Based on the analysis results, we found that South Korea was able to successfully manage the COVID-19 pandemic thanks to the appropriate combination of non-pharmaceutical measures and its advanced public health system. Key factors included the country’s monitoring and surveillance capabilities, extensive and aggressive testing, a particularly innovative contact research system and temporary isolation facilities, collaborations between the government, civil society, and for-profit organizations, transparent government communication, and public health human resource development. 

Our fourth research question concerned the areas in which V4 countries should develop their capabilities and capacities to better manage the COVID-19 pandemic and future pandemics. Based on the experience and lessons learned from the South Korean example, the V4 countries should further develop their monitoring capacities and surveillance systems. Capacity building for the diagnosis of communicable diseases is recommended. It is also advisable to develop further the existing contact research systems and tools. Successful pandemic management also requires the continuous development of human resource capacities and the promotion of international knowledge and information exchange. We recommend the V4 countries engage in research on the treatment of communicable diseases, vaccine development, and establishing collaborations with for-profit and non-governmental organizations. In order to further investigate the topic, it may be helpful to analyze which cultural and social factors in South Korea have helped to combat the pandemic. 

## Figures and Tables

**Figure 1 tropicalmed-06-00201-f001:**
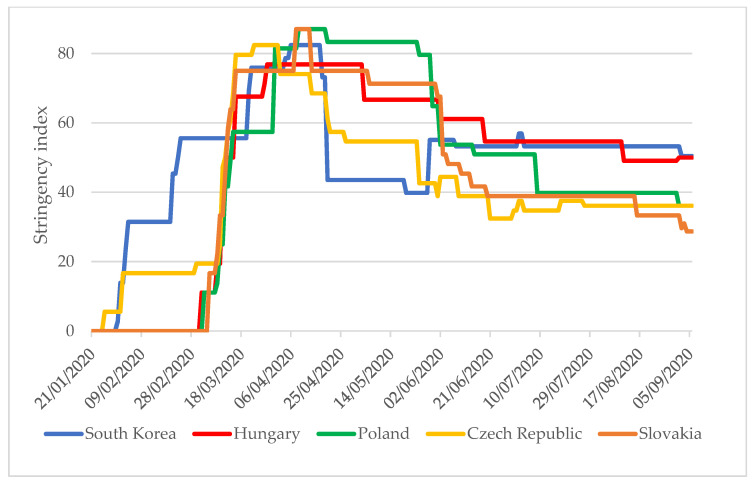
South Korea and V4 countries stringency index values between 1 January 2020 and 6 September 2020 [4].

**Figure 2 tropicalmed-06-00201-f002:**
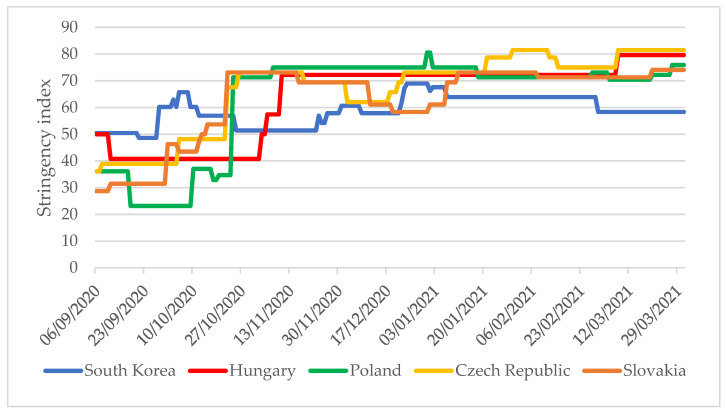
Stringency index values for South Korea and the V4 countries from 6 September 2020 to 31 March 2021 [4].

**Figure 3 tropicalmed-06-00201-f003:**
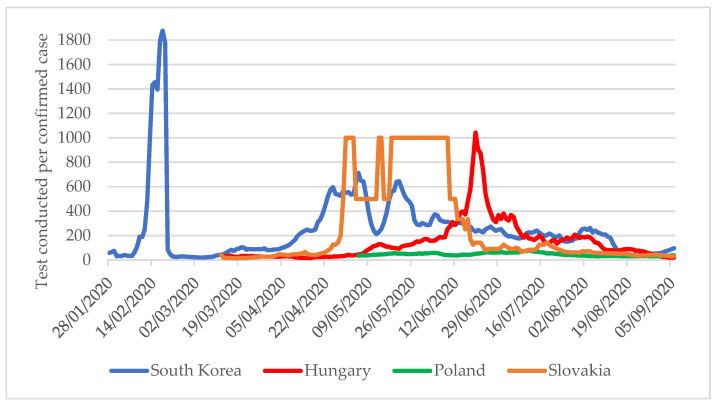
Number of tests per confirmed COVID-19 case in South Korea, Hungary, Slovakia, and Poland between January and 6 September 2020 [4].

**Figure 4 tropicalmed-06-00201-f004:**
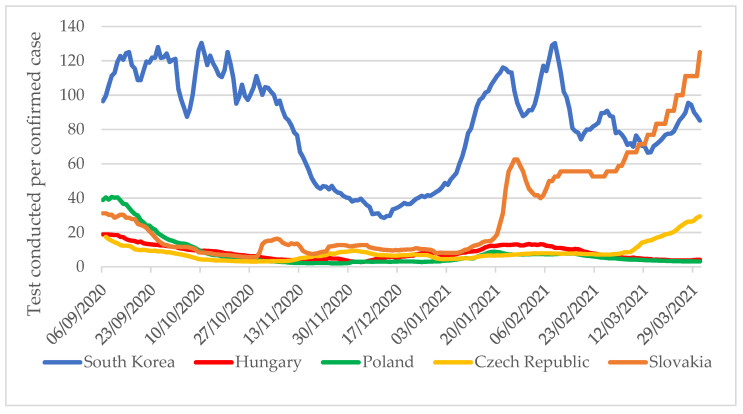
Number of tests per confirmed COVID-19 case in South Korea and V4 countries from 6 September 2020 to 31 March 2021 [4].

**Figure 5 tropicalmed-06-00201-f005:**
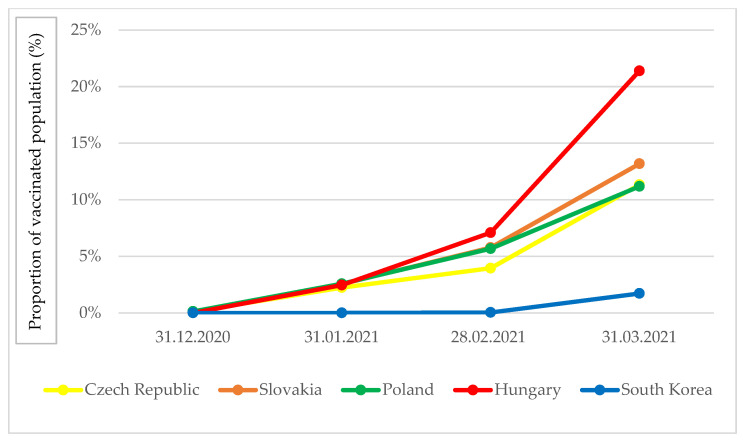
Proportion of populations vaccinated with at least one vaccination dose in South Korea and V4 countries [4].

**Figure 6 tropicalmed-06-00201-f006:**
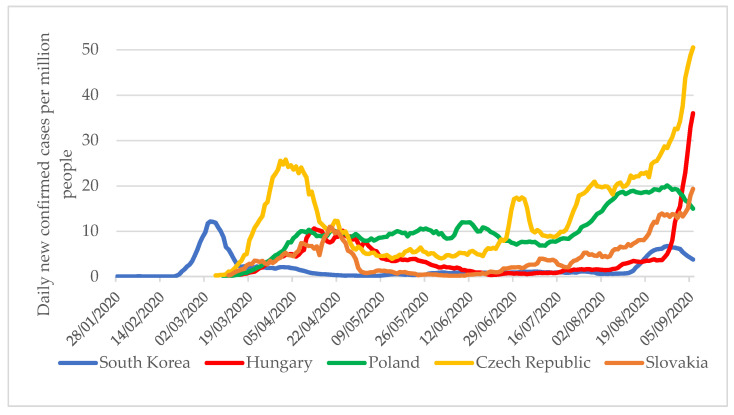
Seven-day moving average of new cases per one million population per day in South Korea and V4 countries between January 2020 and 6 September 2020 [4].

**Figure 7 tropicalmed-06-00201-f007:**
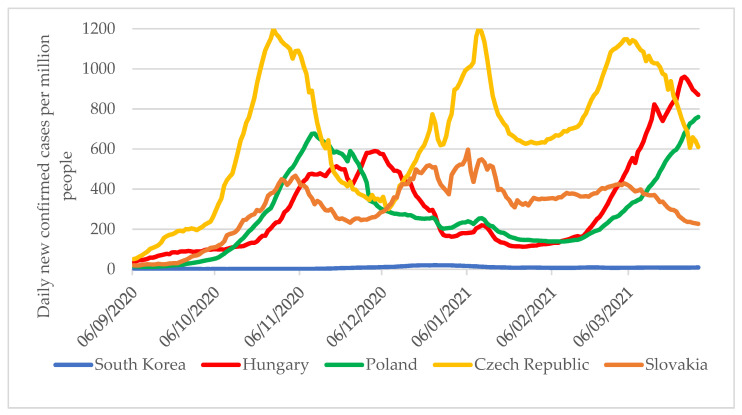
Seven-day moving average of new cases per one million population per day in South Korea and V4 countries from 6 September 2020 to 31 March 2021 [4].

**Figure 8 tropicalmed-06-00201-f008:**
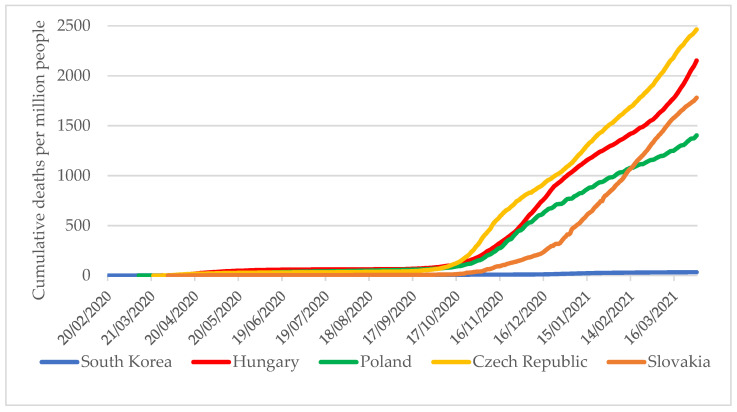
Cumulative numbers of confirmed COVID-19 deaths per one million population in South Korea and V4 countries as of 31 March 2021 [4].

**Figure 9 tropicalmed-06-00201-f009:**
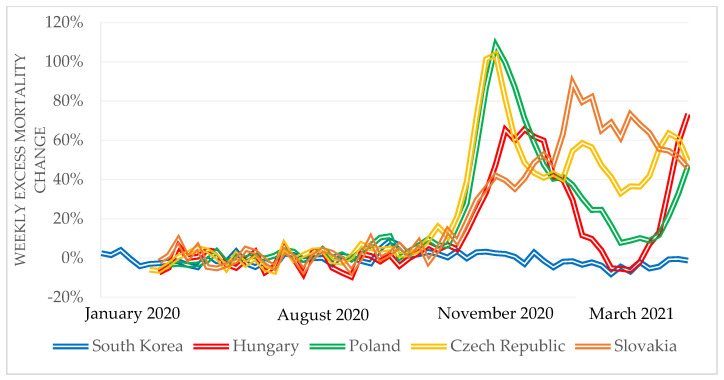
Weekly excess mortality in South Korea and V4 countries during the COVID-19 pandemic compared to the average of the previous five years, January 2020 to 28 March 2021 [4].

**Figure 10 tropicalmed-06-00201-f010:**
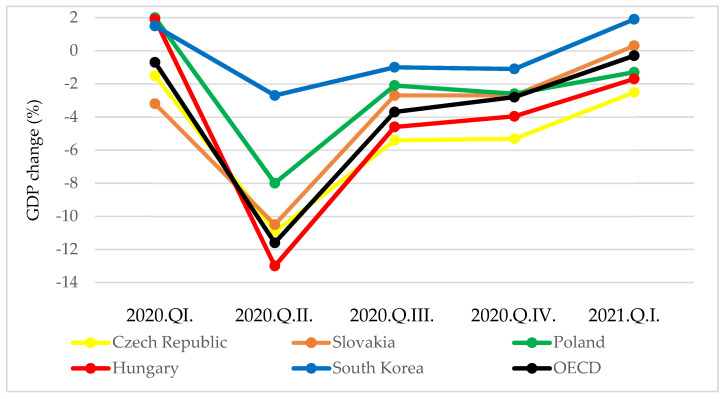
Quarterly GDP changes (%) year-on-year in South Korea and V4 countries [23].

**Figure 11 tropicalmed-06-00201-f011:**
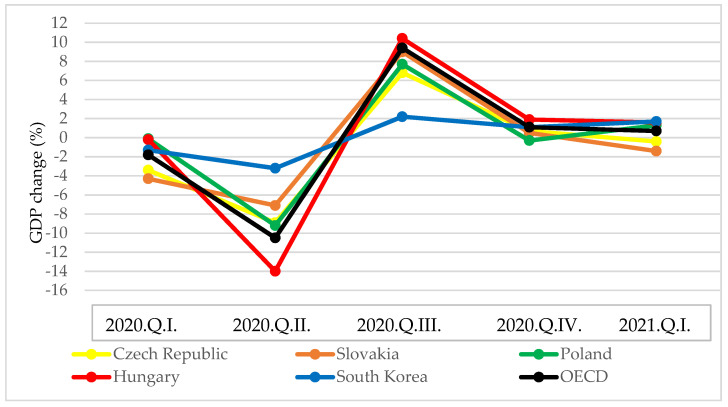
Quarter-on-quarter change in GDP (%) in South Korea and V4 countries [23].

**Table 1 tropicalmed-06-00201-t001:** Start dates of face-covering policies in South Korea and V4 countries [4].

Country	Recommended	Required in Some Public Places	Required Outside the Home All the Time
Czech Republic		19 March 2020	20 November 2020
Slovakia		28 April 2020	15 November 2020
Poland		18 April 2020	
Hungary		28 April 2020	
South Korea	16 March 2020	14 October 2020	

**Table 2 tropicalmed-06-00201-t002:** Start dates of testing policies in South Korea and V4 countries [4].

Country	Targets for Testing Are Individuals with Symptoms + Other Criteria	Testing Targets Any Symptomatic Individuals	Testing Targets Are Both Symptomatic and Asymptomatic Individuals
Czech Republic	10 March 2020	27 March 2020	16 December 2020
Slovakia	2 February 2020	9 July 2020	4 September 2020
Poland	22 March 2020	3 June 2020	
Hungary	29 February2020	14 September 2020	
South Korea			7 February 2020

**Table 3 tropicalmed-06-00201-t003:** Key factors in South Korea’s successful public health policies and pandemic management.

Essential Public Health Operations	Measure/Instrument/Method Used by South Korea	Sources
Monitoring (EPHO 1 + 2)	Apply real-time surveillance systems Simulate outbreaks	[27,28,29,30]
Health protection (EPHO 3)	Establish and implement legislation to support the effective management of health emergencies Develop strategies to reduce the number of physical contacts made and the likelihood of transmitting the infection, adaptable to the pandemic situation	[24,26,31,32]
Health promotion (EPHO 4)	Implement public health programs supporting the development of a healthy environment and lifestyle	[33,34,35,36]
Disease prevention (EPHO 5)	Carry out extensive, aggressive testing Operate an innovative contact research system Establish temporary isolation facilities	[29,32,37,38,39]
Governance (EPHO 6)	Coordinate the responsibilities of national and local organizations responsible for health emergency management	[26,29,32]
Public health workforce (EPHO 7)	Conduct continuous training of public health professionals, supporting their international knowledge exchange Develop health and public health capacities, involving and training volunteers during emergencies	[25,26,29]
Funding (EPHO 8)	Provide the resources needed to operate and develop the public health system	[26,32]
Communication (EPHO 9)	Uphold transparent and wide-ranging communication to all groups of the population	[29,32]
Research (EPHO 10)	Support research into the diagnosis and treatment of communicable diseases Join PPP collaborations with biotechnology companies	[24,26,29,32]

**Table 4 tropicalmed-06-00201-t004:** Levels of trust in the government in South Korea and the V4 countries in 2018 and 2020 (confidence in government—percentage of respondents with yes, %) [42].

Country	2018	2020
Poland	42.7	27.3
Czech Republic	42.1	31.8
South Korea	38.9	44.8
Hungary	38.8	42.8
Slovakia	32.7	30.7

## Data Availability

Data are available in a publicly accessible repository that does not issue DOIs, and publicly available datasets were analyzed in this study. These data can be found here: https://ourworldindata.org/explorers/coronavirus-data-explorer
https://data.oecd.org/gga/trust-in-government.htm. (accessed on 1 August 2021); https://data.oecd.org/gdp/quarterly-gdp.htm
https://data.oecd.org/gga/trust-in-government.htm. (accessed on 1 August 2021); https://data.oecd.org/gga/trust-in-government.htm. (accessed on 1 August 2021).

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
