# Peer review of "Experiences and Lessons Learned from COVID-19 Pandemic Management in South Korea and the V4 Countries"

_tropicalmed, 2021, doi:10.3390/tropicalmed6040201_

Round 1
Reviewer 1 Report
The authors should mention in the introduction part the public health system situation in the world related to this pandemic and also its management. Then the authors should explain the novelty of this study. In addition, I recommend addressing these articles and similar works in reference to the manuscript.
- An overview on the epidemiology and immunology of COVID-19, Journal of Infection and Public Health, 2021
- - The Current Recommended Drugs and Strategies for the Treatment of Coronavirus Disease (COVID-19). therapeutics and Clinical Risk Management
The method section should be written with more details and more complete.
Author Response
Dear Reviewer 1!
Thank you for your valuable comments and suggestions. Based on these, we updated the public health situation in the world in the introduction (line numbers 35-36 and 45-49). Based on the first recommended literature, we supplemented the introduction with the geographical distribution of the pandemic (lines 37-41). At the end of the introduction, we described the novelty of our analysis (63–69).
The methodology chapter has been edited and supplemented in several sections, e.g., literature search methods, databases, keywords, inclusion and exclusion criteria (lines 85-99, 113-114, 204-211, 225-229, and 235-246)). Based on the second recommended literature, we also supplemented the methods (lines 113-114).
Kind regards
The authors
Reviewer 2 Report
Brief Summary
The authors' analysis elaborates on a current and complex topic from an interdisciplinary perspective. The manuscript compares the COVID-19 epidemic management strategies of South Korea and the V4 countries and evaluates their social and economic outcomes. Central to the analysis is why South Korea has suffered fewer social and economic losses than the V4 countries. The strength of the analysis is that it reviews the actions of the governments of the V4 countries and their social and economic outcomes by reviewing the key areas and indicators of pandemic management. The authors also suggest the V4 countries on the areas in which it is expedient to develop their capabilities and capacities, indicating the areas where R&D&I resources should be focused in the future.
The analysis is based on literature and database analysis, the quality of the resources used is adequate. The references are relevant and refer to a significant part of scientific publications and databases published within five years. The data used in the analysis are publicly available, and the data analysis and its results are reproducible. The analysis is structurally, formally proportionate and well-articulated, clear and understandable. The tables and figures used to illustrate the topic under study are easy to review and interpret. The conclusions are supported by an analysis of the data and information described in the results section. Overall, the analysis is high-quality work that meets scientific expectations; I recommend its publication with minor additions.
General concept comments
The authors described in detail the two concepts used in the theoretical background of the analysis. The use of these two concepts is appropriate and fits the subject under study. However, it would be advisable to mention other theoretical concepts when examining the topic and why the authors decided to apply these two concepts. At the end of the analysis, it is recommended to identify further research directions and areas related to the topic.
Specific comments
Line numbers 182-187 It is recommended to indicate the databases in which the authors collected the literature. In addition to the selection criteria in the literature, it is also recommended to describe the exclusion criteria.
Line numbers 257-258. It is recommended to explain in more detail and refer to the literature that the development of testing capacities and extensive testing programs is essential due to the high rate of asymptomatic infections in the COVID-19 epidemic.
Line numbers 421, 431, 442, 451, 477, 483, 492, 500 and 508. In the textual analysis following Table 6, it is also suggested to include the titles of the EPHO numbers for more straightforward interpretation, for example (EPHO 1 + 2) Monitoring, (EPHO 3) Health protection.
Line numbers 521-527. The analysis includes trust in government index values in South Korea and the V4 countries for 2018, which suggests that the level of trust in these governments was almost the same before the pandemic. If the OECD database or other international survey contains data on the COVID-19 pandemic period for these countries, it may be worthwhile to supplement the analysis for 2020 and 2021.
Author Response
Dear Reviewer 2!
Thank you for your valuable comments and suggestions. Based on these, we further developed the methodology chapter, supplemented with additional possible frameworks and justifications for the two selected concepts (lines 105-107).
The methodology chapter has been edited and supplemented in several sections, e.g., literature search methods, databases, keywords, inclusion and exclusion criteria (lines 85-99, 113-114, 204-211, 225-229, and 235-246).
We explained in more detail and referenced the literature that the development of testing capacities and extensive testing programs is essential due to the high rate of asymptomatic infections in the COVID-19 pandemic (lines 328-332).
In the textual analysis following Table 3, we included the titles of the EPHO numbers for more straightforward interpretation, for example (EPHO 1 + 2) Monitoring, (EPHO 3) Health protection (lines 507, 518, 529, 538, 564, 570, 579, 587, 595).
Table 4 and the text of the analysis have been supplemented with the government confidence index values ​​for 2020.
Kind regards
The authors
Reviewer 3 Report
Please consider improving your manuscript based on my notes in the manuscript.

Author Response
Dear Reviewer 3!
Thank you for your valuable comments and suggestions. Based on these, we improved the abstract (lines 11–12) and further developed the introduction based on the proposed literature (lines 45–49 and 63–69).
The methodology chapter has been edited and supplemented in several sections, e.g., literature search methods, databases, keywords, inclusion and exclusion criteria (lines 85-99, 113-114, 204-211, 225-229, and 235-246)).
Table 3 was converted to a graph (lines 377-379).
Health outcomes throughout the text have replaced the term social outcomes.
Tables 4 and 5 were converted to graphs (lines 462-463 and 478-479).
We changed the term concept to the framework (lines 501).
In Table 3, we have improved the classification of PPP collaborations (lines 505).
Kind regards
The authors
Reviewer 4 Report
The authors of this study investigated the public health policies and pandemic management of South Korea and the V4 countries and the social and economic outcomes of the measures. Overall, the authors have presented an interesting study of great public health importance.
Some part of the introduction (description of the steps used in the analysis) should be better placed in the methods section.
Figure 1: Did the authors consider the date of the first identified cases regarding the following finding? “However, while significant restrictions were gradually introduced in South Korea from 5 February, drastic and widespread restrictions began to be applied in the V4 countries only a month later.”
Information about the proportion of new cases related to the population in each country would help the comparison of stringency index.
The decimals should be corrected in Table 3.
“In the fourth quarter, the V4 countries recorded a 2.5-4.5-fold decrease in GDP compared to the previous quarter, as was South Korea.” Please clarify.
A description of confidence index should be added to the Methods.
Author Response
Dear Reviewer 4!
Thank you for your valuable comments and suggestions. We edited the introduction and added certain sections to the methodology chapter (lines 85-99).
When comparing the stringency indexes (Fig. 1 and 2), we supplemented the analysis with the dates of the first identified cases and the daily new confirmed cases per 1 million people (lines 272-276, 288-291 and 298-303).
Table 3 was converted to a graph (lines 377-379).
The text of the analysis for Figure 11 has been clarified.
The methodology chapter has been edited and supplemented in several sections, e.g., literature search methods, databases, keywords, inclusion and exclusion criteria (lines 85-99, 113-114, 204-211, 225-229, and 235-246)).
Kind regards
The authors